# Prices and Trends in FDA-Approved Medications for Sarcomas

**DOI:** 10.3390/cancers16081545

**Published:** 2024-04-18

**Authors:** Caleb Hwang, Mark Agulnik, Brian Schulte

**Affiliations:** 1School of Medicine, University of California, San Francisco, CA 94143, USA; 2Division of Oncology, Keck School of Medicine of USC, Los Angeles, CA 90033, USA; 3Division of Hematology and Oncology, Department of Medicine, University of California, San Francisco, CA 94158, USA; 4Helen Diller Family Comprehensive Cancer Center, University of California, San Francisco, CA 94158, USA

**Keywords:** sarcoma, chemotherapy, progression-free survival, approval, medication, incidence, cost, rare disease, price

## Abstract

**Simple Summary:**

Sarcomas are a group of rare heterogeneous neoplasms of mesenchymal origin. Despite attendant difficulties in study of such disease states, there has been at least one US Federal Drug Administration (FDA) treatment approval for sarcomas in 8 of the last 11 years since 2013. The relative costs, as well as trends in the marketplace, for medications approved for sarcoma have heretofore been unexplored. Given the expansion of medical treatments for sarcoma subtypes in the past decade, it is vital to assess the current status of the landscape to better comprehend achieved successes, challenges, and future directions. Herein, we provide an overview of some of the trends in FDA approvals as well as their associated costs and correlations with incidence as well as outcomes.

**Abstract:**

Sarcomas represent a diverse set of both malignant and benign subtypes consisting of often rare and ultra-rare conditions. Over the course of the last decade, there have been numerous FDA approvals for agents treating various sarcoma subtypes. Given this burgeoning landscape of sarcoma treatments, we seek to review current FDA-approved agents with respect to their rates of incidence, approval rates, and financial costs. We gathered clinical trial data by searching FDA approval announcements from 2013 to 2023. We determined the 30 day and one year cost of therapy for patients of FDA-approved sarcoma treatments in the aforementioned timeframe. From 2013 to 2023, 14 medications have been FDA-approved for sarcoma subtypes. The 30-day dosing prices for these medications range from $11,162.86 to $46,926.00. Since 2013, the rates of approval for sarcoma medications have been higher than in prior decades. Nonetheless, there remains the potential for significant financial toxicity for patients living with sarcoma.

## 1. Introduction

Sarcomas are a rare group of mesenchymal neoplasms that altogether represent 1% of adult malignancies [1]. Historically, they have been divided into soft tissue and bone sarcomas. Within each subdivision exists myriad subtypes, with, for example, soft tissue sarcomas having over 100 distinct histologies [2]. Confounding this issue further is the coexistence of benign entities, such as desmoid fibromatosis, neurofibromas, or tenosynovial giant cell tumor (TGCT). Until the last decade, clinical trials for soft tissue sarcomas generally relied on broad application of generally efficacious medications to this group of neoplasms in aggregate [3,4]. While data from initial studies were able to show general efficacy, an unfortunate consequence was a lack of progress in therapy, with the standard first line treatment for most patients with soft tissue sarcoma remaining a doxorubicin-based therapy [3,5]. Nonetheless, within broader families of soft tissue sarcoma existed some bright spots where molecular underpinning and targeted therapeutics were capable of making significant headway and shifting the landscape of therapy significantly [6]. Over the ensuing years, the field has understandably shifted from wide-spanning application of cytotoxic or nonspecific treatments to more focused and nuanced, and therefore honed and precise, trials aimed at shifting the needle in rare and ultra-rare tumor types [7,8]. This, in conjunction with an inspirational grass roots patient advocacy and expansion in clinical trial networks, has led to FDA approvals of an accelerating number of agents for patients who, previously, had few acceptable options [9,10]. Of note, within these approvals lies the distinction of the FDA’s full approval and accelerated approval processes, in which the latter uses surrogate endpoints such as laboratory measurements or radiographic images in place of direct measures of clinical benefit to considerably shorten the time needed for approval [11].

It is hard to overstate the significance of these developments. With any advances, however, it is nonetheless vital to review the implications of these changes. Therefore, the objective of this study is to review the current environment of FDA-approved agents in sarcoma. Specifically, we seek to qualify the rate of approvals, as well as anticipated costs associated with them, as it may pertain to measured benefits of the treatment, as well as incidence. These could allow for a better understanding of future directions within sarcoma by reflecting on the successes of the past.

## 2. Materials and Methods

### 2.1. Compiling Literature

To assess the body of FDA-approved sarcoma drugs from 2013 to 2023, we first classified the drugs through their class and mechanism of action by searching PubMed for English language studies of these medications. In order to more specifically represent the landscape of approvals in sarcoma, tissue-agnostic therapies were not included in this analysis. It is important to note, however, that patients with select mutations may qualify for these treatments. We also gathered literature that outlined incidence rates of each respective sarcoma indications. Given different incidence rate denominators among the collected articles, we converted all incidence rates to be in terms of per 1,000,000 people. For diseases such as Gastrointestinal Stromal Tumor (GIST) with PDGFRA Exon 18 Mutation, where the literature provided PDGFRA Exon 18 Mutation incidence as a percentage of total GIST cases, we multiplied this given percentage by GIST’s incidence rate. Next, we conducted reviews of the clinical trials that resulted in approvals in the United States for their associated treatments by searching the FDA approval announcements during this time period. Within the compiled sources, we recorded the enrollment periods, recommended dosing regimens, and median progression-free survival (PFS) of these trials (Table 1).

### 2.2. Calculating Treatment Costs

To determine the approximate cost of drugs in United States Dollars (USD) faced by patients, we compiled Average Wholesale Price (AWP) Package Price and AWP Unit Price from the Merative Micromedex RED BOOK of every drug reviewed [40]. To determine the 30-day and one-year cost of therapy for patients, we calculated approximate cost of the drugs by multiplying AWP package price (if the drug is administered intravenously) or AWP unit price (if the drug is administered orally) with the recommended dosing schedule and amounts from the individual trials. Given that these drugs have different lengths for their individual time for AWPs, we applied conversion factors to make all costs equivalent to 30-day treatment cycles. If the drug dose is based on body surface area, such as with intravenous administrations, we multiplied the recommended dose by the average body surface area in the United States (1.7 m^2^) [41]. If the drug dose is based on weight, such as with oral administrations, we multiplied the recommended dose by the average weight in the United States (80 kg) [42].

## 3. Results

### 3.1. Overview of Drugs

FDA approvals for medications occurred in 2013, 2015, 2016, 2019, 2020, 2021, 2022, and 2023. In this 11-year period, the FDA approved 14 agents for subtypes of sarcoma. The FDA withdrew 1 of these 14 drugs in 2020 (olaratumab for advanced soft tissue sarcoma) because of the lack of clinical benefit when compared to standard doxorubicin in the confirmatory phase III trial [43]. Of the 14 agents discussed in this study, 10 are targeted drugs, including kinase inhibitors (themselves representing a varied spectrum of targets), mTOR inhibitors, and methyltransferase inhibitors (Table 2). Of the 14 drugs, 2 are cytotoxic medications: eribulin and trabectedin. These are a microtubule inhibitor and an alkylating agent, respectively (Table 2). Just 1 of the 14 agents, atezolizumab, is an immunotherapeutic, specifically targeting PD-L1 on cancer cells. These 14 drugs utilize nine unique mechanisms of action. 2020 had the highest number of FDA drug approvals for sarcoma, with a total of five, while 2014, 2017, and 2018 had no drug approvals (Figure 1).

### 3.2. Incidence Rates of Sarcoma Subtypes

For malignant sarcoma subtypes, disease incidences ranged from less than 1 to 47 per 1,000,000 people (Table 1). For the benign subtypes, desmoid tumors, tenosynovial giant cell tumor, and neurofibromatosis type 1 (NF1) and symptomatic inoperable plexiform neurofibromas, incidences were 3 to 5, 43, and 737.5 per 1,000,000 people, respectively (Table 1).

### 3.3. Medication Costs with Correlations to Median Progression-Free Survival

To assess medication costs for patients, we calculated the price of 1 month (30 days) and 12 months of usage for each drug following the dosing schedules of their respective clinical trials. A 1-month dosing schedule for eribulin for liposarcoma with prior anthracycline is the least expensive at $11,162.86, while ripretinib for gastrointestinal stromal tumor is the most expensive at $46,926.00 (Figure 2). The medications included in our study had varying levels of efficacy, as determined by measurements such as patients’ median PFS in months in their associated clinical trials. For malignant conditions, these median PFS ranged from 2.6 months for liposarcoma with prior anthracycline to 20.8 months for alveolar soft part sarcoma (Table 1). In addition, for malignant conditions, statistical analyses between individual medications’ median PFS and their monthly costs of treatment in USD revealed a Pearson Correlation Coefficient of 0.45 (Figure 3). Median PFS for the non-malignant conditions discussed in this study, desmoid tumors, tenosynovial giant cell tumor, and NF1 and symptomatic inoperable plexiform neurofibromas were 15.9 months, 16.8 months, and 80.5 months, respectively (Table 1). To better contextualize financial burdens in a typical treatment course, we also calculated the total costs of therapy to patients by multiplying median PFS in months of individual drugs by the price of 30 days of administration. These costs ranged from $29,023.44 for eribulin to $1,961,109.61 for selumetinib (Table 1).

### 3.4. Medication Costs with Correlations to Median Overall Survival

Of the clinical trials discussed in this study, the only one which specifically aimed to compare overall survival to that of standard-of-care therapy in a randomized fashion a priori was by Schöffski et al., in which they found that eribulin’s median overall survival was 13.5 months compared to dacarbazine’s median overall survival of 11.5 months [16].

## 4. Discussion

Herein, we have reviewed the current landscape of FDA-approved therapies for patients with soft tissue neoplasms and tumors. In total, there have been 13 new approvals within the last decade. Interestingly, nine of those approvals have taken place within the last 5 years. Of note, four FDA approvals (38%) occurred in 2020, reflecting that approval rates were not negatively impacted by the onset of the coronavirus disease 2019 (COVID-19) pandemic. Other years that had medication approvals in this timeframe only had either one or two. In addition, it is encouraging to see that the majority of years from 2013 to 2023 had at least one medication approval, reflective of the field’s increasing technological innovation, nuance in treatments, and advocacy group involvement. Furthermore, one of those agents, olaratumab, was withdrawn from the market after failure of a phase 3 randomized controlled trial [4]. Separately, olaratumab and tazemetostat are unique compared to other medications discussed in this study as they were approved under the FDA’s accelerated approval program. Amongst those treatments, the majority have been targeted therapies (77%), while immunotherapy was FDA-approved for one condition, alveolar soft part sarcoma, and two cytotoxic treatments were approved. What is remarkable is that in the decade prior, there was only a single FDA approval within this spectrum of diagnosis, pazopanib for non-adipocytic soft tissue sarcomas based on the PALETTE study [45]. Clearly, these approvals have been the result of dramatic changes in the ability of cooperative groups to leverage their resources and expertise to advance the field and therefore the treatment of many patients with these diseases [46].

At the same time, we must recognize the increasing and at times insurmountable costs associated with administration of these treatments. We have found that the average cost per annum is $310,789.72. The average price per patient per month has likewise reached levels not previously seen, at $28,121.22. It is important to note that such high prices may be partly justified by the nature of studying rare conditions and mobilizing the robust trial network to perform adequate clinical trials. After all, with low incidence, the duration and relative cost per patient will be higher. While beyond the scope of this particular study, it may be wise to create systems by which established metrics such as survival or quality of life are incentivized.

Our data reveal sobering truths, among them that the prices of medications for patients with sarcoma generally exceed 20,000 USD monthly—far in excess of the median income in the United States [47]. This trend has not abated, with more recent approvals commanding increased prices. While these medications have met the threshold as determined by the FDA for approval and marketing, nonetheless, we feel that it is paramount that we be transparent in the cumulative costs associated with their administration. Only through careful analysis and consideration might we develop or determine the best strategies to serve patients, the fundamental stakeholders in the medical system. Of note, the extent of excess medication costs despite increased diversity of options is not exclusive to sarcomas. Previous literature has described prices of brand-name oncology drugs still increasing over time despite more market competition. For example, this has occurred in the fields of non-small cell lung cancer and melanoma, where treatment prices for these cancers have increased without evidence of price competition and despite improved survival rates, respectively [48,49]. However, given sarcomas’ overall relative rarity compared to these cancers, sarcoma drug prices are still overall higher. An additional point to consider with high costs of medications are their increased median progression-free survival compared to those of their respective predecessors. In particular, a number of the drugs discussed in this study, such as ripretinib, have been approved as a result of phase III clinical trials. In the INVICTUS trial, the ripretinib group’s median progression-free survival was 6.3 months compared to the placebo’s median progression-free survival of 1.0 months [32]. Clearly these significant improvements in outcomes come with a premium on financial costs. This is particularly seen for ripretinib, where its 1-month dosing is the costliest of all discussed medications at nearly $47,000. However, whether these high premiums are completely justified because of these increases in PFS is still a source of contention between manufacturers, clinicians, and their patients.

Additional analyses within our study have determined that there is no correlation, within sarcoma, between measures of efficacy (as measured by median PFS) and pricing. As such, it may be appropriate to speculate that pricing is therefore an independent variable. Based on available data, there was no correlation in this small sample size that could likewise be found between proposed mechanism, duration of treatment, and pricing. We readily admit that, insofar as this is an observational appraisal of the landscape of pricing of therapy, that there may be some unknown factor weighing on this variable—it is just not readily obvious by our estimation.

More recent advances by creation of novel pharmacy strategies have been a bright spot from a pricing standpoint, reducing the cost of generic medications such as imatinib mesylate by factors of over 100 [50]. As more antineoplastic and other medications become available as a generic, there may continue to be expansion of such programs, although the market for medications approved for rare conditions may be substantially smaller.

Despite the many hurdles encountered in the production and study of these medications, the overarching trend of an increased rate of approvals is encouraging. Increased availability of clinical trials and approval of novel agents for rare and ultra-rare conditions, of which soft tissue neoplasms represent a small proportion, has led to improved outcomes for patients with these conditions and revolutionized therapy for many.

## 5. Conclusions

Since 2013, the expansion of treatments available for sarcoma subtypes in the United States has been a heartening development, reflecting the field’s shift from non-specific, though generally efficacious, treatments to more nuanced targeted therapies. Therefore, these medications will better allow clinicians to optimize their patients’ outcomes. Still, given soft tissue tumors’ and neoplasms’ rarity, it is unfortunately not surprising that many of their risks are not just medical side effects but also financial toxicities. Though there has been greater availability of generic medications and expansion of clinical trials for novel therapeutics, in the long term, it remains to be seen how effective costs can be contained to an affordable and palatable cost for patients. Overall, our work shows there remain significant hurdles to address for patients living with sarcoma to receive effective and affordable treatments for their conditions, but that the expansive growth in knowledge and understanding of this broad group of cancers offers hope for the future.

## Figures and Tables

**Figure 1 cancers-16-01545-f001:**
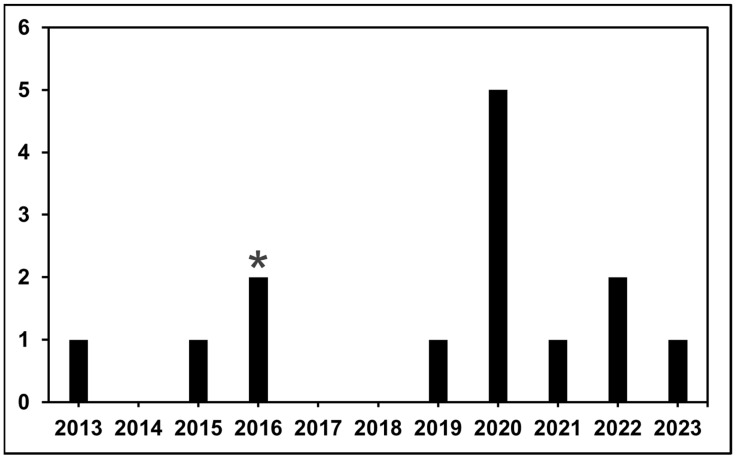
Bar graph of the number of medication approvals by the FDA by year since 2013. The asterisk represents olaratumab (indicated for advanced soft tissue sarcoma), which was FDA-approved in 2016 and withdrawn in 2020.

**Figure 2 cancers-16-01545-f002:**
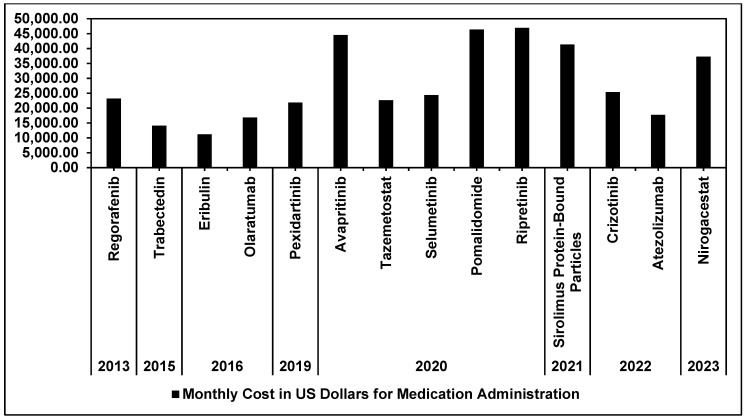
Bar graph of FDA-approved drugs arranged by year of approval (*x*-axis) and monthly cost of each drug in US Dollars (*y*-axis).

**Figure 3 cancers-16-01545-f003:**
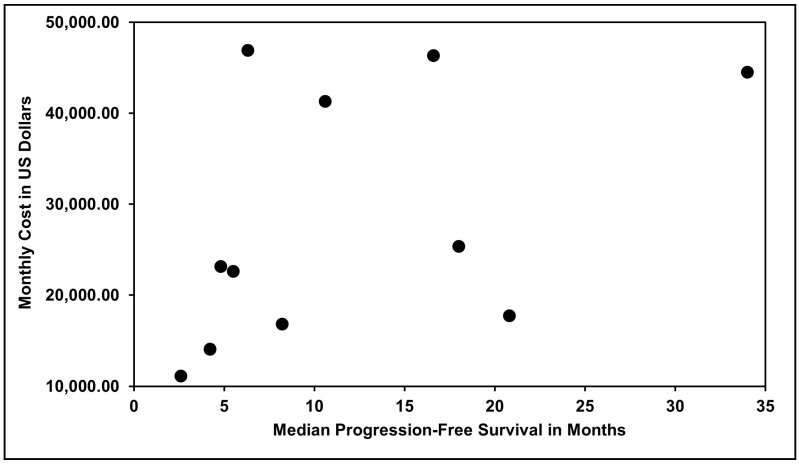
Scatter plot of sarcoma drugs’ median progression-free survival in months versus the monthly cost of the drug in USD. Nirogacestat, selumetinib, and pexidartinib were excluded from this scatter plot as their respective indications (desmoid tumors, neurofibromatosis type 1 and symptomatic inoperable plexiform neurofibromas, and tenosynovial giant cell tumor) are benign. The Pearson Correlation Coefficient and *p*-value for the datapoints is 0.45 and 0.164, respectively.

**Table 1 cancers-16-01545-t001:** Tabulation of sarcoma indications with respective drug name, year of FDA approval, incidence rates per 1,000,000 people, AWP price in USD, days for AWP, price for 30 days of drug, price for 12 months of drug, median PFS in months, and total cost of therapy to patient.

Indication	Medication	Year of FDA Approval	Incidence of Disease per 1,000,000 People	AWP in USD	Time for AWP (Days)	Price for 30 Days of Drug (1 Month) in USD	Price for 12 Months of Drug in USD	Median PFS in Months	Total Cost of Therapy to Patient (Median PFS × Price for 30 Days of Drug)
Third line treatment Gastrointestinal Stromal Tumor	Regorafenib [12]	2013	10–15 [13]	21,628.32	28	23,173.20	444,925.32	4.8	111,231.36
Liposarcoma and Leiomyosarcoma	Trabectedin [14]	2015	Liposarcoma: 7 Leiomyosarcoma: 5 [15]	9860.57	21	14,086.53	78,884.52	4.2	59,163.43
Liposarcoma who have received prior anthracycline	Eribulin [16]	2016	Liposarcoma: 7 [15]	7814.00	21	11,162.86	93,768.24	2.6	29,023.44
Soft Tissue Sarcoma nthracycline-containing is appropriate	Olaratumab [17]	2016	47 [18]	11,788.80	21	16,841.14	169,758,72	8.2	138,097.35
Tenosynovial Giant Cell Tumor	Pexidartinib [19,20]	2019	43 [21]	21,865.2	30	21,865.20	503,774.16	16.8	367,335.36
Gastrointestinal Stromal Tumor with PDGFRA Exon 18 Mutation	Avapritinib [22]	2020	1.6 [23]	44,533.20	30	44,533.20	534,398.40	34	1,514,128.80
Epithelioid Sarcoma	Tazemetostat [24]	2020	0.5 [25]	22,620.00	30	22,620.00	255,542.40	5.5	124,410.00
NF1 and Symptomatic Inoperable Plexiform Neurofibromas	Selumetinib [26,27]	2020	NF1/NF2: 737 [28,29]	24,361.61	30	24,361.61	331,312.32	80.5	1,961,109.61
Kaposi Sarcoma	Pomalidomide [30]	2020	6 [31]	43,280.00	28	46,371.43	556,465.56	16.6	769,765.74
Gastrointestinal Stromal Tumor	Ripretinib [32]	2020	10–15 [33]	46,926.00	30	46,926.00	563,112.00	6.3	295,633.80
Perivascular Epithelioid Cell Tumor	Sirolimus Protein-Bound Particles [8]	2021	0.3 [34]	28,940.80	21	41,344.00	347,292.00	10.6	438,246.40
Inflammatory Myofibroblastic Tumor	Crizotinib [35,36]	2022	Less than 1 [37]	25,389.00	30	25,389.00	304,674.72	18 [38]	457,002.00
Alveolar Soft Part Sarcoma	Atezolizumab [7]	2022	Less than 1 [7]	12,416.00	21	17,737.14	149,000.16	20.8	482,002.56
Desmoid Tumors	Nirogacestat [39]	2023	3 to 5 [39]	34,800.00	28	37,285.71	447,428.52	15.9	592,842.79

**Table 2 cancers-16-01545-t002:** Sarcoma medications categorized by drug class, further divided into the mechanisms of action, their individual names, and the number of drugs in each class.

Drug Classification	Mechanism of Action	Drug Name	Number of Drugs
Cytotoxic	Microtubule-Dynamics Inhibitor [16]	Eribulin	1
	Alkylating Agent [44]	Trabectedin	1
Targeted	Angiogenesis Inhibitor [30]	Pomalidomide	1
	EZH2 Inhibitor [24]	Tazemetostat	1
	Kinase Inhibitor [12,19,22,26,33,36]	Regorafinib ^1^ Pexidartinib ^2^ Avapritinib ^3^ Selumetinib ^4^ Ripretinib ^5^ Crizotinib ^6^	6
	PDGFR-α Target [17]	Olaratumab	1
	mTORC1 Inhibitor [8]	Sirolimus Protein-Bound Particles	1
	γ-Secretase Inhibitor [39]	Nirogacestat	1
Immunotherapy	PD-L1 Target [7]	Atezolizumab	1

^1^ Regorafinib targets VEGFR1-3, TEK, KIT, RET, RAF1, BRAF, and BRAF^V600E^. ^2^ Pexidartinib targets CSF1R, KIT, and FLT3-internal tandem duplication. ^3^ Avapritinib targets KIT and PDGFR-α. ^4^ Selumetinib targets MEK. ^5^ Ripretinib targets KIT and PDGFR-α. ^6^ Crizotinib targets ALK.

## Data Availability

The data can be shared up on request.

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
