# Peer review of "Prices and Trends in FDA-Approved Medications for Sarcomas"

_cancers, 2024, doi:10.3390/cancers16081545_

Round 1
Reviewer 1 Report
Comments and Suggestions for Authors
Manuscript entitled” Prices and Trends in FDA Approved Medications for Sarcomas” has been written well by the author. However, few points have been mentioned below which need to be corrected.
Line no.: 70-72: How incident rate was converted? Please provide detail information about formula/calculation factor used.
Line no. 72-76: Provide references of clinical trial reviewed.
Line 88 and 90: please provide references for the calculations.
Line 94 is contradictor with line no. 13. Please explain in detail.
As per the manuscript, cost was determined for last ten year, and usually drug manufacturing companies have patents, and they sell their invented medicines at higher prices. I recommend that author should add comparison of drug treatment cost with patent (within 12-20 years) and generic (whose patent is over). Also, should add efficacy comparison with old and new regimes for better clarity for readers.
Reviewer 2 Report
Comments and Suggestions for Authors
Thank you for the opportunity to review your paper.
This article discusses trends in FDA approval for sarcoma research. The perspective it offers is significant for the advancement of treatments in rare cancers. Therefore, I believe this paper holds importance for publication. Below, I provide some minor points for consideration.
1 please change the numeral letters in the first sentence to capital. (e.g., Line 101, 103)
2 please discuss about other basket type trials, such as treatments for NTRK mutation, BRAF mutation, nivolumab.
These kind treatments can be used patients with specific gene mutations and it might be case with sarcoma treatment.
3 I think indication “table 1” might be “table2 (line 99,101,112)
4 you can provide p value in Figure 3.
Comments on the Quality of English Language
please change the numeral letters in the first sentence to capital. (e.g., Line 101, 103)
Reviewer 3 Report
Comments and Suggestions for Authors
Thank you for this overview on costs for sarcoma treatments. I have some minor comments and on major isssue.
Figure 1 shows the approval of 14 drugs, of which one was withdrawn in 2020. In which year was this drug approved? Would it be helpful for the reader to mark this drug in the graphic? Or otherwise only show the 13 drugs you review in the rest of the paper.
Line 108-113: You mention that disease incidences are shown in table 1, however in table 1 are medication categories by drug class and no incidence rates? Do you mean table 2?
Line 127: do you mean figure 2 instead of figure 3?
Line 180-181: I do not understand “It is important to note that such high process may be partly justified by the….” do you mean such high prices?
Line 195-196: is this only the fact for non-small cell lung cancer? What about oncology drugs in general?
Line 231-232: you state that the new medications allow to optimize patient’s quality of life. But you have not at all described if any of these newer drug does increase quality of life. Therefore, such a “conclusion” is not justified.
My major issue is:
Discussion:
Why not calculate costs per median overall survival instead of PFS. Or at least both. Aren’t there any studies on incremental cost effectiveness ratios for sarcoma drugs? See for example: https://pubmed.ncbi.nlm.nih.gov/?term=ICER+sarcoma
For Ripretinib see: https://pubmed.ncbi.nlm.nih.gov/34938653/
These results should be discussed.
Comments on the Quality of English Languagenone
Round 2
Reviewer 3 Report
Comments and Suggestions for Authors
Thank you for implementing my suggestions.